# Fabrication and Characterization of Ag-Graphene Nanocomposites and Investigation of Their Cytotoxic, Antifungal and Photocatalytic Potential

**DOI:** 10.3390/molecules28104139

**Published:** 2023-05-17

**Authors:** Sidra Batool Malik, Asma Gul, Javed Iqbal Saggu, Banzeer Ahsan Abbasi, Beenish Azad, Javed Iqbal, Mohsin Kazi, Wadie Chalgham, Seyed Arshia Mirjafari Firoozabadi

**Affiliations:** 1Department of Biological Sciences, International Islamic University, Islamabad 44000, Pakistan; sidra.phdbt64@iiu.edu.pk (S.B.M.); beenish.phdbt76@iiu.edu.pk (B.A.); 2Department of Physics, Quaid-i-Azam University, Islamabad 45320, Pakistan; javed.saggu@qau.edu.pk; 3Department of Botany, Rawalpindi Women University, 6th Road, Satellite Town, Rawalpindi 46300, Pakistan; 42Department of Botany, Bacha Khan University, Charsadda 24420, Pakistan; javed89qau@gmail.com; 5Department of Pharmaceutics, College of Pharmacy, King Saud University, P.O. Box 2457, Riyadh 11451, Saudi Arabia; mkazi@ksu.edu.sa; 6Department of Mechanical and Aerospace Engineering, University of California, Los Angeles, CA 90095, USA; wadie.chalgham@ucla.edu; 7Department of Microbiology, Immunology, and Molecular Genetics, University of California, Los Angeles, CA 90095, USA; arshimi@ucla.edu

**Keywords:** silver nanoparticles, graphene, antifungal, cell viability, nanocomposites, degradation

## Abstract

In the present study, we aimed to synthesize (Ag)_1−x_(GNPs)_x_ nanocomposites in variable ratios (25% GNPs–Ag, 50% GNPs–Ag, and 75% GNPs–Ag) via an ex situ approach to investigate the incremental effects of GNPs (graphene nanoparticles) on AgNPs (silver nanoparticles). The prepared nanocomposites were successfully characterized using different microscopic and spectroscopic techniques, including X-ray diffraction (XRD), Fourier transform infrared (FTIR) spectroscopy, ultraviolet spectroscopy, and Raman spectroscopic analysis. For the evaluation of morphological aspects, shape, and percentage elemental composition, SEM and EDX analyses were employed. The bioactivities of the synthesized nanocomposites were briefly investigated. The antifungal activity of (Ag)_1−x_(GNPs)_x_ nanocomposites was reported to be 25% for AgNPs and 66.25% using 50% GNPs–Ag against *Alternaria alternata*. The synthesized nanocomposites were further evaluated for cytotoxic potential against U87 cancer cell lines with improved results (for pure AgNPs IC_50_: ~150 µg/mL, for 50% GNPs–Ag IC_50_: ~12.5 µg/mL). The photocatalytic properties of the nanocomposites were determined against the toxic dye Congo red, and the percentage degradation was recorded as 38.35% for AgNPs and 98.7% for 50% GNPs–Ag. Hence, from the results, it is concluded that silver nanoparticles with carbon derivatives (graphene) have strong anticancer and antifungal properties. Dye degradation strongly confirmed the photocatalytic potential of Ag-graphene nanocomposites in the removal of toxicity present in organic water pollutants.

## 1. Introduction

In the modern era, nanoscience and nanotechnology have been utilized in various aspects, including health, drug synthesis, and ecosystem improvement [1,2,3,4,5]. Currently, diverse nanomaterials, e.g., Ag-CuO, graphene oxide, Fe_3_O_4_/SiO_2_-HAp, Fe_2_O_3_, CaWO_4_, and citric acid dendrimers, have been investigated for various applications, including medical and environmental applications [6,7,8,9,10,11,12]. Noble metallic nanostructures (Au, Ag, Pd, Pt, etc.) have been examined for a wide range of applications, including photography, catalytic activities, molecular detection, diagnosis, antimicrobial and cytotoxic assays, wastewater treatment, textiles, surgical tools, treating HIV, utilization in pharmaceutics, chemical and biosensing, and storage batteries [13,14,15,16,17]. Among all these noble metallic nanoparticles, silver nanoparticles have gained much attention from researchers because of their high-class properties and multiple applications (as catalysts and biomedical agents) in chemical reactions [18]. AgNPs have been exploited in textiles, cosmetics, the food industry, and biomedicine. Among all their applications in the biomedical field, they have earned much strength, especially as antimicrobial agents, coated material in medical devices, and media in chemotherapy. The antibacterial activities of AgNPs have been broadly studied, but researchers have also focused on biomedical applications against fungal, viral, amoebic, cancer, angiogenic, and inflammatory activities [19]. AgNPs inhibited the fungus *Candida albicans* [20]. Researchers have also investigated the antiviral activity against the hepatitis B virus (HBV) via AgNPs [21] and antiamoebic activity [22]. It has also been reported that cytotoxicity generated by AgNPs can be an active candidate against cancer. AgNPs were successfully tested against breast, liver, and lung malignancies and have potential as drug carriers during chemotherapy. AgNPs inhibit HepG2 cell lines with an IC50 of 2.8 ppm (µg/mL) compared to normal cell lines with an IC50 of 121.7 ppm [23]. AgNPs cause toxicity in human cells at low concentrations [24]. For this purpose, different organic substances are used to control the discharge of silver ions [25]. Graphene with a single-layered sp2 hybridized structure packed in a 2D honeycomb lattice structure is considered one of the most utilized carbon derivatives [26]. By reducing graphene oxide (GO), graphene is obtained [27]. Graphene allows the healthy combination of metallic nanoparticles with diverse properties to yield novel and improved nanomaterials. GO is commonly utilized to blend with AgNPs [28,29,30]. Fungi are now considered health threats to animals, plants, and ecosystems [31]. Mycotoxicosis, a condition caused by ingesting mycotoxins produced on foodstuffs, leads to acute poisoning, liver failure, and cancer [32]. Considering yield losses, *Alternaria* is one of the most devastating documented saprophytic and pathogenic genera [33]. Within this genus, *Alternaria alternata* has been reported to produce more than 30 mycotoxins [34]. Fungicides are nonbiodegradable, and their accumulation in soil, water, and plants imposes adverse effects on the ecosystem. Nanotechnology has offered a new hope to cope with these issues by exploiting the use of nanostructured materials in multiple ways [35]. A study [36] explored the activity of GO-AgNPs nanocomposite against the phytopathogen *Fusarium graminearum* both in vitro and in vivo and showed three to seven times more activity than pure AgNPs and GO, respectively. Cancer is a multidimensional disease, extremely variable in its appearance, expansion, and consequence, caused by a complex blend of genetic and environmental aspects. Many drugs against cancer are incapable of targeting the site properly and fail to drop the pharmacological effect without causing unwanted injury to healthy cells and tissues [37,38,39]. Nanotechnology offers techniques to treat cancer by passing nanodrugs that cross biological barriers to targeted sites directly without harming neighboring cells [40]. The exclusive physicochemical features of metallic nanoparticles, such as a high surface-to-volume ratio, extensive optical characteristics, surface functionalization, and ease of synthesis, have opened new gates for cancer therapy [41,42,43]. Carbon-derived nanomaterials, e.g., graphene, have been investigated extensively to assess cellular toxicity in multiple cancer cells [44], such as human alveolar basal epithelial (A549), breast [45], and ovarian cancer cells [27], owing to their distinctive traits. Silver nanoparticles have also been investigated against human lung cancer cells [46]. In addition, environmental issues are most alarming; hence, the handling of various organic contaminants (pollutants) in wastewater is on the current side of environmental issues [47]. Numerous methods, including chemical and physical methods, have been utilized to eradicate organic waste. Physical approaches, including ultrafiltration, chemical agent coagulation, activated carbon adsorption, ion exchange, and reverse osmosis, have also been utilized to deal with wastewater. Nanotechnology has paved the way to hack this alarming concern by applying nanomaterials under controlled conditions [48]. In the present study, we investigated the production of silver and graphene nanocomposites through an ex situ approach for the first time with varying ratios of GNPs (graphene nanoparticles). The synthesized Ag/GNPs were characterized by adopting XRD techniques for crystalline structure and nanocomposite size and FTIR for the analysis of functional groups and stretching modes. The morphological appearance and elemental composition of the nanocomposites were evaluated using SEM and EDX, respectively. The incremental effects of GNPs on AgNPs (silver nanoparticles) were investigated for the first time through cytotoxic analysis against glioma U87 cancer cell lines as antifungal agents against *A. alternata* (saprophytic fungi) and as photocatalysts in the degradation of Congo red (toxic organic pollutant produced by industries as waste) under visible light.

## 2. Results

### 2.1. Characterization of Synthesized Nanocomposites

Aqueous suspensions of AgNPs and (Ag)_1−x_(GNPs)_x_ were utilized to record the UV-visible spectra. The data were recorded in the 200–700 nm range through a spectrophotometer (Figure 1). The presence of AgNPs in the graphene nanoparticles was confirmed by the absorption peaks at 411 nm. This particular range of absorbance spectra shows the surface plasmon effect of AgNPs, confirming the attachment and presence of AgNPs on the graphene nanostructure due to surface plasmon resonance (SPR) of silver nanoparticles, which is fairly good, as reported in earlier data [49]. It is observed that (Ag)_1−x_(GNPs)_x_ nanocomposites show slightly different spectra compared to AgNPs. This may be due to the insertion of GNPs into AgNPs. The bond energy between C–C atoms (π–π*) present in GNPs is remarkably transformed after the addition of AgNPs; the reason for this phenomenon is the strong conjugated π-bond system due to the π–π stacking interaction between GNPs and AgNPs [50,51]. The Eg values (energy band gaps) of AgNPs and (Ag)_1−x_(GNPs)_x_ nanocomposites are calculated using Tauc’s relation given below:(αhυ) n = A (hυ − Eg)
where A is a constant, hυ is the photon energy, Eg = hc/λ, λ is the wavelength, α is the absorption coefficient, and n is an integer; its value is measured by the direct/indirect bandgap. For direct bandgap transitions at n = 2, the Eg obtained for AgNPs is 3.86 eV, which is fairly good in comparison with previously observed values, i.e., 3.73 eV 60; a reduction in the bandgap with the addition of GNPs is observed, hence shrinking the energy band gaps to 2.36 eV, 2.95 eV, and 3.08 eV for the 75% GNPs–Ag nanocomposite, 50% GNPs–Ag nanocomposite, and 25% GNPs–Ag nanocomposite, respectively (Figure 1). The XRD spectra of the synthesized (Ag)_1−x_(GNPs)_x_ nanocomposites are shown in Figure 2. The XRD analysis confirmed the crystalline nature of AgNPs. The resulting Bragg peaks were in accordance with single and pure phase face-centered cubic AgNPs with JCPDS card no. 96-110-0137. The XRD patterns showed varying distinct major diffraction peaks with 2Ø values of 38.11, 44.30, 64.44, and 77.40, which corresponded to the (111), (200), (220), and (311) Bragg reflections, respectively. The XRD pattern obtained for GNPs showed major diffraction peaks at 2Ø 26.54 (002) and 54.66 (100), confirming the hexagonal structure with JCPDS card no 96-901-1578. The diffraction peaks observed in the (Ag)_1−x_(GNPs)_x_ nanocomposites (Figure 2 insets) are at 2Ø 26.54 (002), 38.11 (111), 44.30 (200), 64.44 (220) and 77.40 (311). The peak at 2Ø 26.54 (002) in Ag/GNPs nanocomposite became intense as the % age of GNPs increased, hence confirming the anchoring of GNPs onto the AgNP surface. No additional XRD peaks were observed, indicating the successful synthesis of nanocomposites. The average size of AgNPs is calculated as ~24 nm, GNPs as ~32 nm, and (Ag)_1−x_(GNPs)_x_ nanocomposites as 34 nm. Our XRD configuration for the described nanocomposites is consistent with earlier studies [52,53,54]. The functional groups present in the (Ag)_1−x_(GNPs)_x_ nanocomposites were confirmed by exploiting an FTIR analysis to obtain the desired spectra (Figure 3). The major peaks observed for GNPs are at 860, 1533, and 3577 cm^−1^ (Figure 3a), and the peaks are attributed to epoxy, ether/peroxidase groups, C=O carbonyl vibrations, and H-OH stretching. Peaks at 2007 and 2146 cm^−1^ are attributed to the stretching of the C-H vibrational modes of methylene [55]. The FTIR of AgNPs had peaks at 772 cm^−1^ (C-H stretching), 981 cm^−1^ (due to C-O stretching of esters), 1357 cm^−1^ (N=O nitro group vibrational mode), 1572 cm^−1^ (due to C=O aldehydes and ketones), and 2214 cm^−1^ (C-OH stretching), while a broad peak at 3358 cm^−1^ and a sharp peak at 3649 cm^−1^ were due to H-OH stretching [56]. The FTIR of graphene/Ag nanocomposites (25% GNPs–Ag, 50% GNPs–Ag, and 75% GNPs–Ag) showed prominent peaks at 610, 825, 1751, 2348, 3176, and 3489 cm^−1^ (Figure 3b). New peaks appeared at 825 cm^−1^ (C-H stretching) with 25% GNPs–Ag, 1751 cm^−1^ (C=O) with 50% GNPs–Ag, and 1763 cm^−1^ (C=O) with 75% GNPs–Ag [57]. Morphological analysis was performed by scanning electron microscopy (SEM). The images obtained for pure AgNPs and GNPs are shown in Figure 4d,e. The images for silver and graphene nanocomposites (25% GNPs–Ag, 50% GNPs–Ag, and 75% GNPs–Ag) are shown in Figure 4a–c. Silver and graphene particles are visible with fine marginal ends, and AgNPs are rounded in shape with an average particle size nearly equal to 24 nm. GNPs appear flat and leafy in shape, suggesting a platelet-like structure with fine margins and a well-composed shape with an average particle size of 31 nm. SEM obtained for (Ag)_1−x_(GNPs)_x_ nanocomposites reflects unique features, as graphene nanoparticles hold silver nanoparticles on their surface, confirming the successful synthesis of nanocomposites. It is also clear from Figure 4 that as the ratio of GNPs increases, there is more intensification of the nanocomposites while preserving the identity of the AgNPs on their surfaces. Hence, graphene nanoplatelets are well decorated by AgNPs, both small and large particles. The average nanocomposite size was calculated to be 29 to 34 nm. The nanocomposite size becomes larger because of the interaction between the parent nanoparticles, as this phenomenon has already been discussed in previous studies [58,59]. The investigation of the percentage elemental composition of the nanocomposites was performed by EDX analysis, obtained in the form of a spectrum (Figure 5). The percentage of elemental composition was also indicated without any other impurity in the prepared nanocomposites.

### 2.2. Biological Assays

#### 2.2.1. Antifungal Activity

The inhibitory effect of (Ag)_1−x_(GNPs)_x_ nanocomposites at a concentration of 1 mg/mL was analyzed on SDA media. The maximum growth inhibition (GI) was observed with 50% GNPs–Ag (66.25% GI), followed by 75% GNPs–Ag (52.5% GI), 25% GNPs–Ag (50% GI), and AgNPs (25% GI). It is clear that the addition of graphene nanoplatelets onto the silver nanoparticle surface enhanced the inhibition of mycelium growth, as reported in previous studies that GO-AgNPs, when tested against the phytopathogen *F. graminearum* in vitro and in vivo, are three to seven times more potent than pure Ag and GO nanoparticles [36]. Mycelium growth was studied after examining the growth diameter from day one to seven (Figure 6) and it was observed that the rate of mycelium growth was slower in fungi treated with Ag/graphene nanocomposites than in those treated with pure AgNPs. The observed mycelium growth diameter values are shown in Table 1. Hence, it is concluded that (Ag)_1−x_(GNPs)_x_ nanocomposites not only inhibit mycelium growth inhibition but also slow the growth rate because of the synergistic effect of graphene-based NPs after physical injury to cells followed by the release of reactive oxygen species, as studied earlier [36]. Cell death occurs after the strong association of nanoparticles with the microbial membrane [59]. The ratio between the surface area and the volume is increased; hence, antimicrobial activity is increased because of the small size of nanoparticles [60].

The results indicated that damage to the mycelium and spores was associated with the mechanism of the (Ag)_1−x_(GNPs)_x_ nanocomposites. Oxidative stress on microbial cells commonly occurs after treatment with various nanomaterials, including metallic oxide nanoparticles and carbon-based nanomaterials [61,62,63,64]. Reactive oxygen species (ROS) are a family of reactive oxidants with very short life spans [63]. ROS are considered harmless offshoots obtained after aerobic metabolism, but cells in biological systems are responsible for excessive ROS production under hostile conditions [62]. Unnecessary ROS production is responsible for the induction of injury to multiple cells. Structures, including lipids, proteins, and DNA, influence signaling pathways, leading to cell death [64]. Hence, it is concluded that the establishment of a connection with fungi, (Ag)_1−x_(GNPs)_x_ nanocomposites not only induces morphological injury but is also responsible for ROS production, which continues to inhibit fungal spores, finally leading to cell death (Figure 7). The aforementioned phenomenon was also reported in a study in which GO-AgNPs nanocomposites showed excessive antimicrobial action at an extremely low concentration (9.37 μg/mL) compared to pure AgNPs (12.45 μg/mL) and GO nanosheets (250 μg/mL). The activity of both GO sheets and AgNPs plays a vibrant role, not only encouraging physical damage in biological systems but also inducing oxidative stress in cells [36].

#### 2.2.2. Cytotoxicity of Ag-Graphene Nanocomposites

To assess the cytotoxicity of the (Ag)_1−x_(GNPs)_x_ nanocomposites, a cell viability assay was performed against U87 cell lines, and IC50 values were recorded against each nanocomposite and pure nanoparticles. It was observed that the IC50 value of AgNPs was higher than that of (Ag)_1−x_(GNPs)_x_ nanocomposites, suggesting that Ag/graphene both have synergistic effects in killing cancerous cells, as reported in earlier studies [65]. The range of concentration of synthesized nanocomposites was from 3.15 to 200 µg/mL (Figure 8). With the increase in concentration, cell death increases, therefore decreasing the percentage of cell viability. This phenomenon was observed in earlier findings; as the concentration of the drug increased, the percentage viability also decreased [66]. In this study, the incremental effect of graphene on AgNPs was studied; hence, 25% GNPs–Ag, 50% GNPs–Ag, and 75% GNPs–Ag were synthesized (as discussed earlier in the methodology) and applied in a cell viability assay against U87 cell lines.

Nanoparticles are reported to induce cytotoxic effects on human cells by adopting multiple cellular mechanistic approaches: (a) by taking up free nanoparticles that lead to defective DNA replication, (b) by producing free radicals and reactive oxygen species (ROS), and (c) by inducing stress on the cell membrane, which deforms its overall structure followed by cell damage or death [67,68]. The possible mechanism of action of (Ag)_1−x_(GNPs)_x_ against cancerous cells is shown in Figure 9. The altered mitochondrial membrane after ROS production triggers caspase 3 (a protein involved in the destruction of cell organelles and DNA fragmentation), followed by apoptosis, thus arresting the cell cycle. Once apoptosis is initiated, it leads to the activation of cellular pathways, including the activation of P53 proteins, and an increase in P53 proteins leads to cell death and destruction of the nucleus [69].

### 2.3. Evaluation of Photocatalytic Performance

The photocatalytic activity of the (Ag)_1−x_(GNPs)_x_ nanocomposite for the degradation of Congo red was achieved under visible light. Prior to the photodegradation experiment, adsorption equilibrium was achieved by stirring the solutions in the dark for 30 min. It was clear that in the presence of AgNPs-GNPs nanocomposites, the dye degraded to 98.7% after exposure to visible light within 75 min (Figure 10). (Ag)_1−x_(GNPs)_x_ nanocomposites were investigated under visible light irradiation utilizing a xenon lamp of 500 W to assess the photocatalytic activity against the anionic toxic organic dye. The results were obtained in the form of absorbance spectral curves (Figure 11). The degradation of the dye increased with time (Figure 12); hence, it is emphatically proven that when there is an increase in exposure time under visible light, the dye moves more toward discoloration. The percentage photodegradation with different graphene ratios and changes in rate reaction are shown in Table 2. Hence, it is concluded that 50% GNPs–Ag showed the maximum photodegradation of CR, and it can be depicted that AgNPs, in combination with GNPs, effectively made healthy interactions with each other and served as conduction support for AgNPs [70]. Pseudofirst-order kinetics is applied for the photocatalytic activity of AgNPs and Ag-graphene nanocomposites by utilizing the expression below.
ln(Co/C) = kt

The rate constants obtained after linearly fitting the obtained data are 0.589, 0.849, 0.936, and 0.860 min^−1^ for AgNPs, 25% GNPs–Ag, 50% GNPs–Ag, and 75% GNPs–Ag nanocomposites, respectively, and the rate constant changes with time (Figure 13). The photocatalytic activity of Congo red under visible light irradiation involves the release of superoxide radicals (O_2_), hydroxyl radicals (OH), and hydrogen peroxide (H_2_O_2_). ROS participate in triggering photodegradation activity. These are produced by the reduction of oxygen and the oxidation of water photocatalytically [71], which is clearly shown in Figure 14. When the dye is exposed to visible light, thus acting as a sensitizer and exciting electrons to the graphene nanoplatelet (electron acceptor) surface, it undergoes self-degradation by the absorbed O_2_ [72]. These electrons from the graphene surface move to the AgNPs and result in the retardation of the combining process of electrons and dyes. It was reported in the literature that graphene contains high charge carrier mobility, hence significantly enhancing the charge transfer and separation of electrons generated by photocatalysis. During exposure to visible light, the formation of free radicals (OH) in solution promotes the dye degradation process more easily [73].

## 3. Discussion

Nanotechnology has been employed in many fields of life, especially as an active candidate in the medical field. Nanoparticles have been employed for the last few decades in wastewater treatment and in the medical field [1,2,3]. To date, many nanomaterials have been investigated for their effective roles in biomedical and environmental applications [6,8,9,10,11]. Metallic nanoparticles have been extensively researched in many fields, including medicine and wastewater treatment. Among metallic nanostructures, silver nanoparticles have gained a special place in the medical field, ranging from antimicrobial to anticancer applications [19]. Because of certain minor cytotoxic effects produced by AgNPs, different organic nanostructures are attached to AgNPs. Graphene is considered one of the best options to overcome silver nanoparticle cytotoxicity [24]. Graphene metallic nanocomposites are in the limelight in the biomedical field (enhanced antibacterial activity) because of their unique properties [74]. Therefore, we investigated the role of graphene in assisting AgNPs against *A*. *alternata*, glioma cancer cell lines, and Congo red. For this purpose, we synthesized AgNPs and graphene nanocomposites by an ex situ approach for the first time with changing ratios of graphene (25% GNPs–Ag, 50% GNPs–Ag, and 75% GNPs–Ag). Previously, different methods have been employed for the synthesis of nanocomposites. Synthesized nanocomposites were characterized through different techniques to confirm their healthy and fruitful synthesis. Figure 15 shows the whole schematic approach of (Ag)_1−x_(GNPs)_x_ nanocomposites from synthesis to biomedical applications, along with their characterization. XRD analysis confirmed the successful synthesis of Ag/G nanocomposites. Optical analysis and band gap calculation confirmed the synthesis of silver graphene nanocomposites. The silver band gap shrinks because of the addition of graphene [51]. Functional group analysis of the synthesized (Ag)_1−x_(GNPs)_x_ was investigated, and the observed data were consistent with previously reported studies showing stretching and vibrational modes of C=O and C=H [57]. The morphological structure of (Ag)_1−x_(GNPs)_x_ was verified through SEM analysis. The average nanocomposite size calculated ranged from 29 to 34 nm. An increase in the nanocomposite size was observed compared with the parent nanoparticles (AgNPs 24 nm, GNPs 31 nm). This was due to the interaction between silver and graphene NPs, as reported in a previous study [59]. EDX analysis confirmed the purity of the nanocomposites.

Fungal diseases affect more than one billion people with serious infections [75]. To control these fungal diseases, different methods have been employed, but pathogens have developed resistance to conventional methods. Nanotechnology offers a new way to tackle these alarming microbial infections present in both plants and animals. The synergistic outcome of GO-AgNPs causes physical injury after the production of reactive oxygen species. G-TiO2 nanocomposite-coated cotton showed remarkable activity against bacteria (*Escherichia coli* and *Staphylococcus aureus*) and fungi (*Candida albicans*) [76]. G was merged with TiO2 nanoparticles to expedite effective bacterial growth inhibition by developing a fruitful connection between nanocomposites and microbial cells [77]. In the present study, we evaluated the antifungal activity of (Ag)_1−x_(GNPs)_x_ against *A. alternata.* It was observed that with the increase in the concentration of the nanocomposites, mycelium growth was inhibited. The gradual increase of GNPs to AgNPs also made inhibition fast. It was previously investigated that after treatment of *Mucor racemosus* with a graphene oxide-borneol (GOB) nanocomposite, long-term antifungal effects were reported, in which fallen spores did not germinate even after five days [78]. When incubation time increases, growth inhibition also increases [77].

Cancer is now a vast and worldwide medical threat to humans. Many factors are involved in this complex disease, including both genetic and environmental factors. Many treatments and procedures have been developed to cure or stop its progression to advanced stages [37,38,39]. The IC50 value recorded for AgNPs was higher than that for Ag-graphene nanocomposites, but among GNPs–Ag nanocomposites, 50% GNPs–Ag showed the best result, with an IC50 of 12.5 µg/mL and a dose concentration of 200 µg/mL. The IC50 values recorded for 25% GNPs–Ag and 75% GNPs–Ag were 140 µg/mL and 90 µg/mL, respectively. Hence, it is emphatically concluded that this was due to the fruitful interaction of AgNPs with GNPs, as has already been reported in previous research [79]. Waste products from industries are always a continuous threat to humans, plants, and aquatic life. Millions of tons of contaminated waste from industries utilizing dyes have been dumped into rivers and ponds. Various conventional methods have been applied for the eradication of this marine pollution. Nanostructured materials are now in the limelight for the treatment of contaminated water from industries. In the present study, we applied synthesized nanocomposites against Congo red. Previous studies have reported that, because of plasmon resonance, AgNPs have the ability to excite light absorption capacity [76]. It was observed that with the addition of GNPs to AgNPs in the calculated ratio, the degradation was enhanced up to 98.7%. This incremental effect is due to the addition of GNPs, as previously reported in many studies [80]. The rate constant also increases with time because some special features of the Ag-graphene nanocomposite serve as good catalysts for the degradation of organic dyes, i.e., strong adsorption capacity, healthy π-π interactions with the aromatic structure of the dye, ample photosensitization of electrons, and low electron recombination processes [81]. It has also been reported that the dye structure has a special role in evaluating the validity of the nanocatalyst, which attacks the functional group to disturb the aromatic structure of the dye [82,83]. Figure 15 shows the whole schematic approach of (Ag)_1−x_(GNPs)_x_ nanocomposites from synthesis to biomedical applications, along with their characterization.

## 4. Materials and Methods

### 4.1. Materials

Silver nanoparticles (99.99%; Guangzhou Hongwu Material Technology, Guangzhou China), graphene nanoparticles (100%; Knano, Xiamen, China), deionized (DI) water, Congo red, Dulbecco’s modified Eagle’s medium (DMEM), fetal bovine serum (FBS), Pen-Strep, trypsin, phosphate buffer saline (PBS), formaldehyde, crystal violet, acetic acid, and dimethyl sulfoxide (DMSO) were used as chemical materials (99.98%; Gibco Chemicals, New York, NY, USA) for the cytotoxicity assay, and Sabouraud dextrose agar (SDA) media (99.99%; Merck, Rahway, NJ, USA) was used for antifungal activity. All chemicals were purchased and utilized as acknowledged without any further purification.

### 4.2. Synthesis of (Ag)_1−x_(GNPs)_x_ Nanocomposites

(Ag)_1−x_(GNPs)_x_ nanocomposites with different weight ratios, such as 25% (x = 0.25), 50% (x = 0.50), and 75% (x = 0.75), of GNPs were prepared via an ex situ approach. Samples with 25% GNPs were prepared by the mentioned method: 25% GNPs and 75% AgNPs were dispersed in DI water and ultrasonicated for approximately 30 min. Furthermore, the GNPs–Ag aqueous solution was stirred at 500 rpm for half an hour at room temperature. The obtained nanocomposite was dried at 80 °C for almost 16 h in an oven and finally ground to a fine powder. Nanocomposites with 50% and 75% GNPs were synthesized by means of a similar approach as discussed above for 25% GNPs–Ag.

### 4.3. General Characterizations

#### 4.3.1. UV-Vis Analysis

The band gap and optical studies of the prepared nanocomposites were evaluated using spectrophotometry. The absorbance spectra were obtained by an Evolution 300 UV-Vis spectrophotometer (Shimadzu, Tokyo, Japan).

#### 4.3.2. XRD

XRD was utilized to examine the crystalline phase of the prepared samples. The prepared nanostructured size was calculated by the following Scherrer formula:D = 0:9ʎ = β cos θ

The XRD of the synthesized samples was carried out with the help of a PANalytical Empyrean diffractometer (PANalytical XRD (Almelo, The Netherlands)).

#### 4.3.3. Functional Group Analysis (FTIR)

By exploiting the technique of KBr pellets, the functional groups present in silver and graphene nanocomposites were assessed. Ten milligrams of the sample in powder form was compressed in 100 mg of KBr pellet to make translucent disks of the sample. These samples were investigated by FTIR spectroscopy [a resolution of 4 cm^−1^, scan ranging from 400 to 38.00 cm^−1^]. FTIR spectra were taken using a Tensor 27 FT-IR spectrometer (Bruker Corporation, Billerica, MA, USA).

#### 4.3.4. Morphological and EDX Analysis

The elemental percentage present in silver and graphene nanocomposites was confirmed using energy-dispersive X-ray spectroscopy (EDX) analysis. Morphological studies were investigated using scanning electron microscopy. The sonication of the samples was performed for approximately 5 min to produce a suspension of the (Ag)_1−x_(GNPs)_x_ nanocomposites in distilled water. Then, a drop of the sample was placed on carbon-coated conductive tape, and the sample was placed under a lamp to dry. A VEGA3 TESCAN instrument was used for analysis.

### 4.4. Biological Assays

#### 4.4.1. Antifungal Assay

Well-preserved cultures of *Alternaria alternata* (*A*. *alternata*) (accession no. MH553296) were allowed to grow on Sabouraud dextrose agar (SDA) media at 25 ± 1 °C for up to 7 days. To evaluate the antifungal activity of AgNPs and GNPs–Ag nanocomposites, a poisoned food technique was adopted. To carry out this activity, SDA media was modified with a concentration of (Ag)_1−x_(GNPs)_x_ nanocomposites (1 mg/mL). A cork borer was used to pick inoculum disks of *A. alternata* (4 mm) positioned in the middle of nanocomposite-modified SDA plates. SDA without nanoparticles was used as a positive control. After inoculation, the petri dishes were incubated at 26 ± 1 for up to 7 days, and the growth inhibition was calculated by the formula below:Percentage Growth Inhibition = (C − T)/C × 100
where C is the growth in positive control plates and T is the growth in treated plates.

#### 4.4.2. Cytotoxicity Assay

The cytotoxic activity of the synthesized (Ag)_1−x_(GNPs)_x_ nanocomposites was evaluated against the commercial cell line U87 (malignant glioma) obtained from Khyber Medical University Peshawar, Peshawar, Pakistan. By adopting an in vitro cell growth inhibitory assay, U87 cells were retrieved, and 1 × 10^4^ cells were seeded in each well of a 96-well plate. The cells were then nourished with DMEM-F12 media with 10% FBS and 1% Pen-Strep and incubated for 24 h in a humidified CO_2_ environment. Cells were critically monitored for growth and infection for 24 h at 37 °C. After 24 h, cells were inoculated/loaded with various doses of (Ag)_1−x_(GNPs)_x_ nanocomposites, AgNPs, and GNPs (0–200 μg/mL) and incubated again for 48 h. Following the incubation period, the cells were washed with PBS, fixed with 4% formalin solution, and incubated for 10 min at room temperature. A crystal violet assay was then immediately performed, where fixed cells were stained with 0.1% crystal violet dye solution, and then excess dye was washed away with PBS. The plate was air-dried, and the amount of dye (crystal violet) absorbed by fixed cells was detected by measuring the absorbance via an Eliza elx 800 at 630 nm. Finally, the IC50 values of the (Ag)_1−x_(GNPs)_x_ nanocomposites, AgNPs, and GNPs were computed and graphed in Microsoft Origin 9.5 for glioma U87 cell lines.

### 4.5. Measurement of Photocatalytic Properties

The photodegradation performances of all samples of the (Ag)_1−x_(GNPs)_x_ nanocomposites were evaluated at room temperature using visible light illumination, and a xenon bulb of 500 Watt was used as the light source. The dye solution was placed directly under a light source at a 15 cm distance. To assess the photocatalytic activity, 40 mg of catalyst was utilized with 100 mL of distilled water as the initial volume, and a 3 ppm aqueous solution of Congo red (CR) was used in all experiments. The reaction mixture was brought to adsorption/desorption equilibrium with the help of magnetic stirring for 30 min in the dark. After removing the photocatalyst, approximately 4 mL of the test solution was taken at regular time intervals (15 min). The collected solution was centrifuged at 4000 rpm for approximately 10 min. The absorbance spectra of CR were recorded by a UV-visible spectrophotometer. The % degradation was calculated using the equation below:% Degradation = (1 − C_t_/C_o_) × 100

## 5. Conclusions

The current study described the synthesis of (Ag)_1−x_(GNPs)_x_ via a simple ex situ approach. The physicochemical characteristics of the nanocomposites were evaluated using different microscopic and spectroscopic techniques, which confirmed the synthesis of the nanocomposites. The biological potential of the prepared nanocomposites was tested against *A. alternata* and U87 glioma cancer cell lines and showed significant potential against these assays. Further nanocomposites proved themselves as active candidates for the photodegradation of Congo red. These nanocomposites can be further exploited for different in vivo studies.

## Figures and Tables

**Figure 1 molecules-28-04139-f001:**
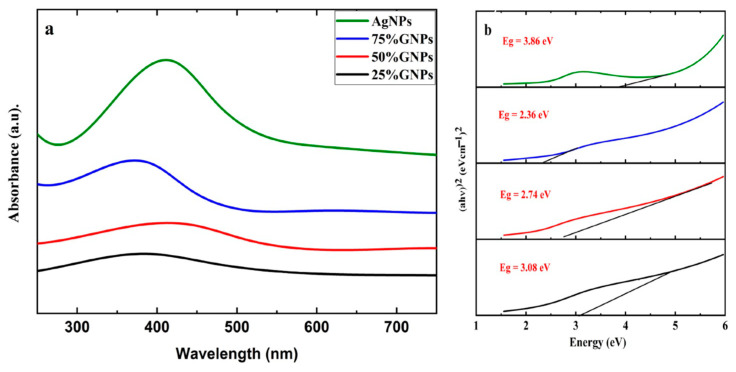
UV-Vis analysis of (**a**) AgNPs and GNP–Ag nanocomposites and (**b**) calculated energy bandgaps of AgNPs and Ag–GNPs nanocomposites.

**Figure 2 molecules-28-04139-f002:**
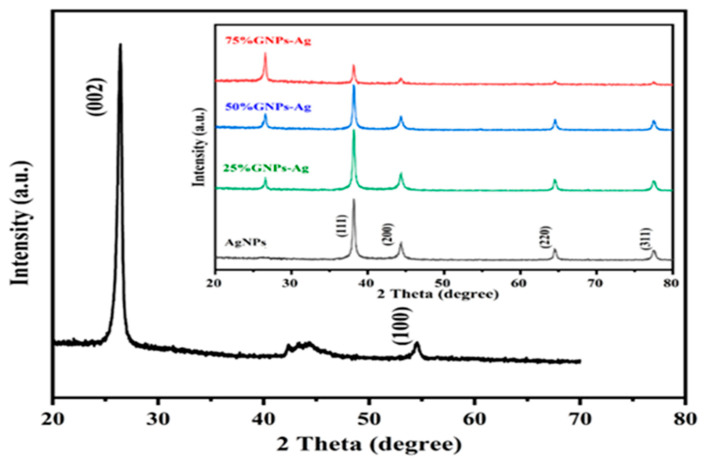
XRD spectral analysis of GNPs, AgNPs, and Ag/G nanocomposites with varying ratios of GNPs (inset).

**Figure 3 molecules-28-04139-f003:**
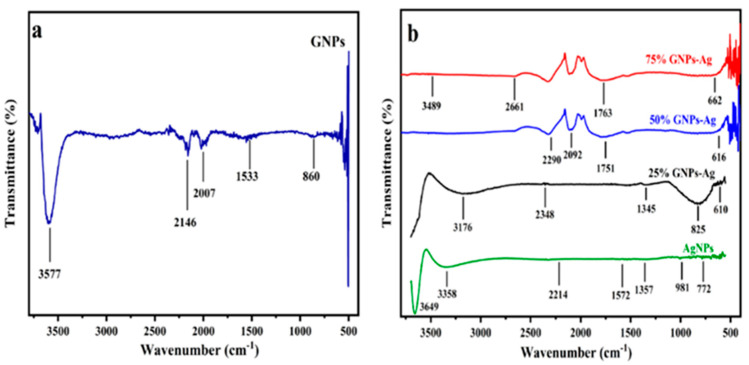
(**a**) FTIR spectrum of GNPs and (**b**) AgNPs and (Ag)_1−x_(GNPs)_x_ nanocomposites.

**Figure 4 molecules-28-04139-f004:**
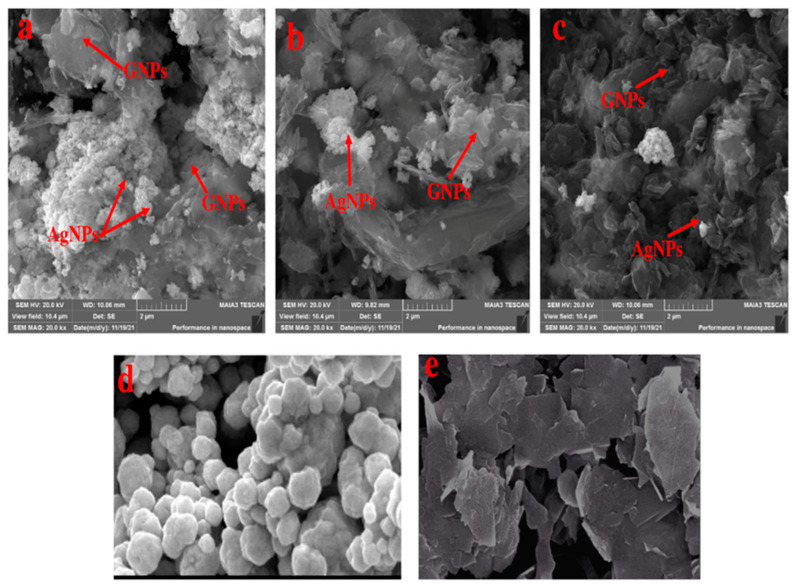
SEM analysis of synthesized nanostructures (**a**) 25% GNPs–Ag, (**b**) 50% GNPs–Ag, (**c**) 75% GNPs–Ag, (**d**) AgNPs, and (**e**) GNPs.

**Figure 5 molecules-28-04139-f005:**
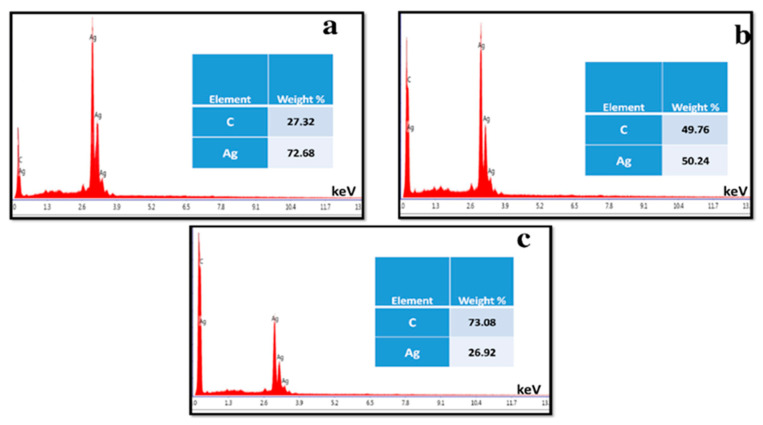
EDX analysis for percentage composition of elements present in (Ag)_1−x_(GNPs)_x_ nanocomposites (**a**–**c**).

**Figure 6 molecules-28-04139-f006:**
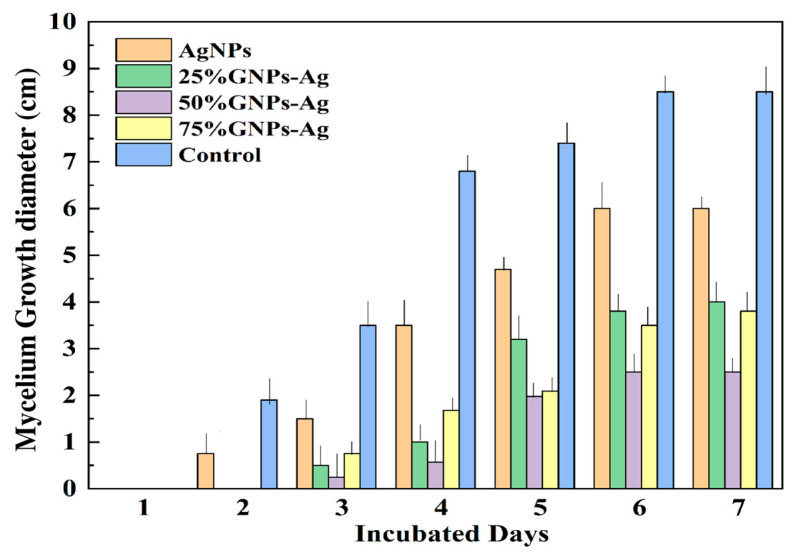
Inhibitory effect of Ag/GNPs on mycelium growth from days 1 to 7 of incubation.

**Figure 7 molecules-28-04139-f007:**
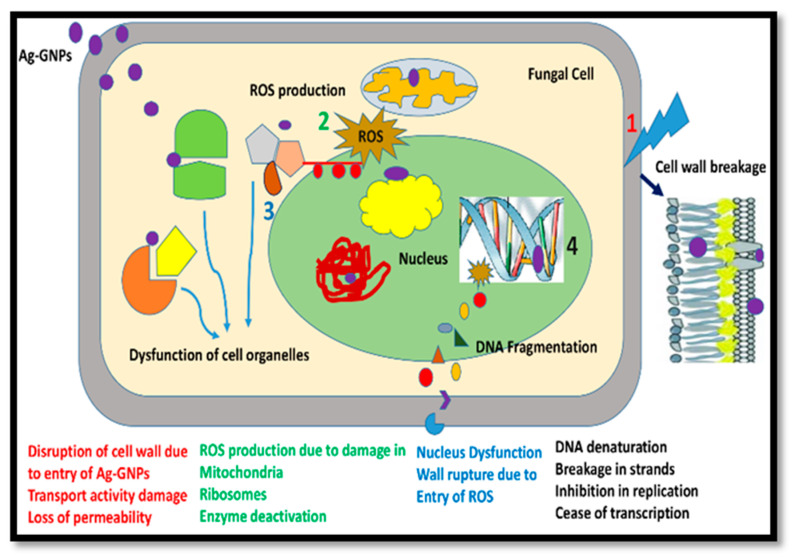
Mechanism of action of Ag-graphene nanocomposites in fungal cell death after the production of ROS. (1) Cell wall breakage (2) ROS production (3) Dysfunction of cell organelles (4) DNA fragmentation.

**Figure 8 molecules-28-04139-f008:**
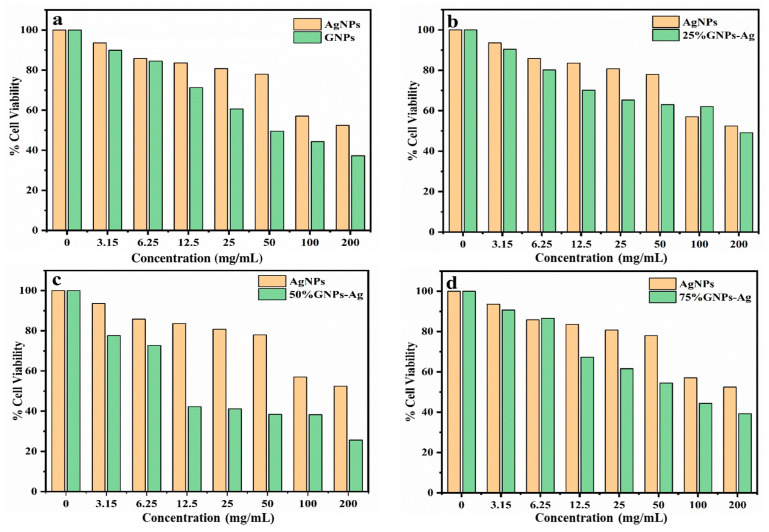
(**a**) Inhibitory effect of AgNPs and GNPs on glioma U87 cell lines. (**b**–**d**) Cytotoxicity of (Ag)_1−x_(GNPs)_x_ nanocomposites against U87 cell lines with incremental effects of GNPs (25%, 50%, and 75%) on AgNPs.

**Figure 9 molecules-28-04139-f009:**
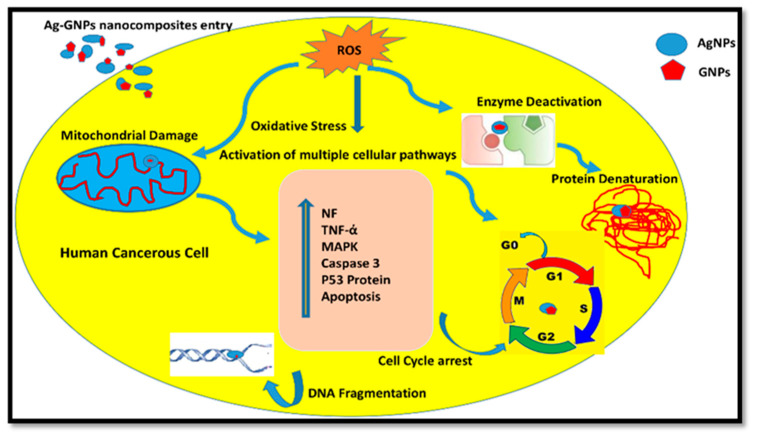
Possible mechanism of action of (Ag)_1−x_(GNPs)_x_ against human cancerous cells.

**Figure 10 molecules-28-04139-f010:**
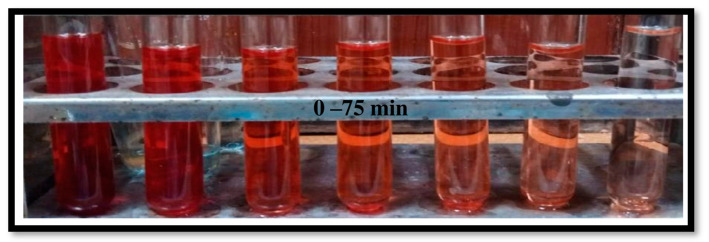
Decolorization of Congo red from 0 to 75 min under visible light irradiation.

**Figure 11 molecules-28-04139-f011:**
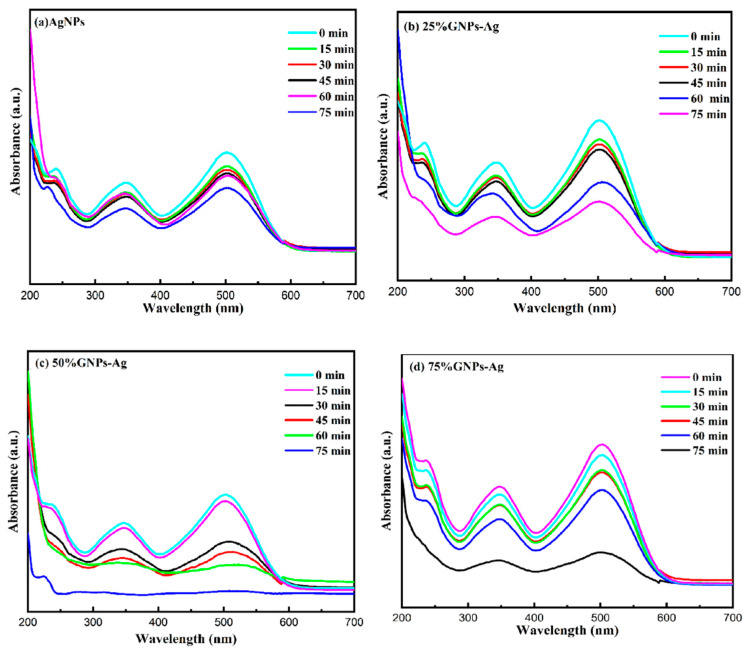
(**a**) Visible light-driven photodegradation of Congo red via AgNPs. (**b**–**d**) (Ag)_1−x_(GNPs)_x_ nanocomposites.

**Figure 12 molecules-28-04139-f012:**
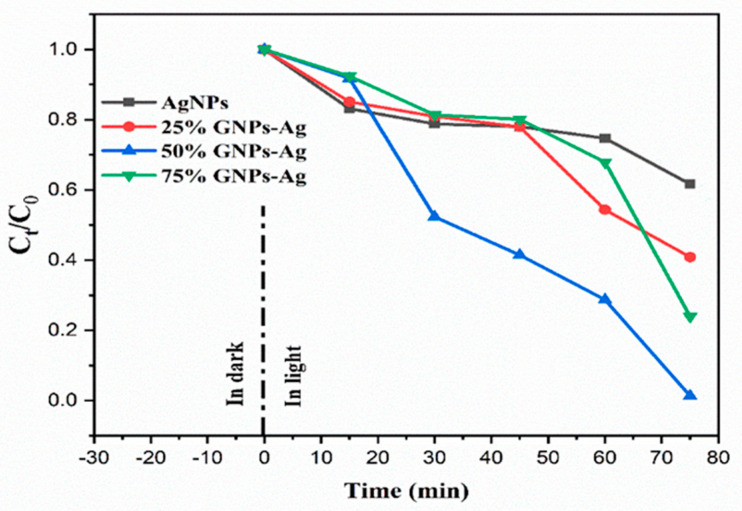
Change in the concentration of Congo red after the addition of photocatalysts (AgNPs, 25% GNPs—Ag, 50% GNPs—Ag, and 75% GNPs—Ag).

**Figure 13 molecules-28-04139-f013:**
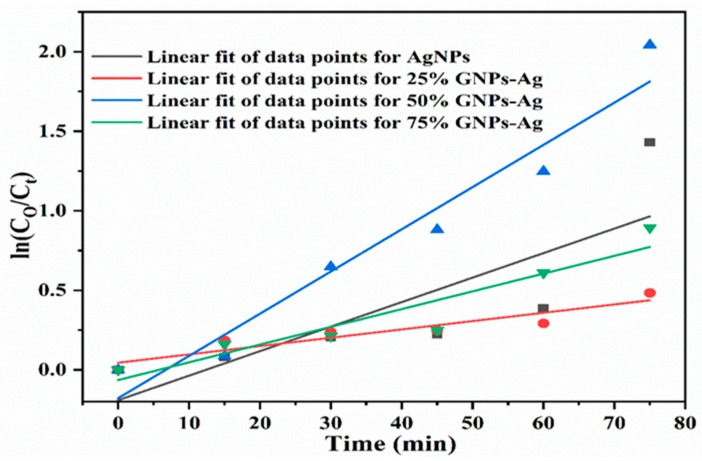
Pseudofirst-order kinetics of catalyst (Ag)_1−x_(GNPs)_x_ against CR.

**Figure 14 molecules-28-04139-f014:**
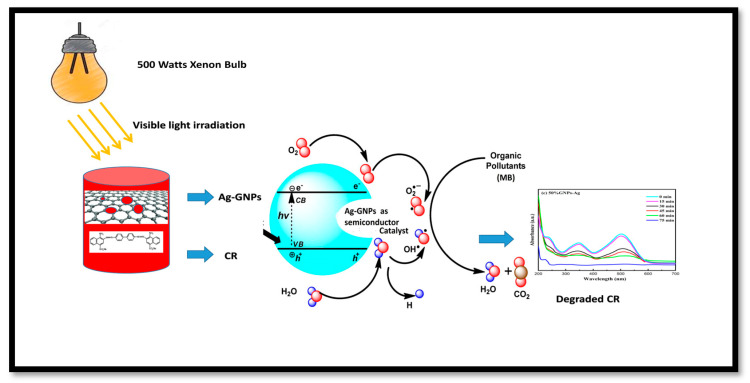
Schematic illustration of CR with (Ag)_1−x_(GNPs)_x_ nanocomposites.

**Figure 15 molecules-28-04139-f015:**
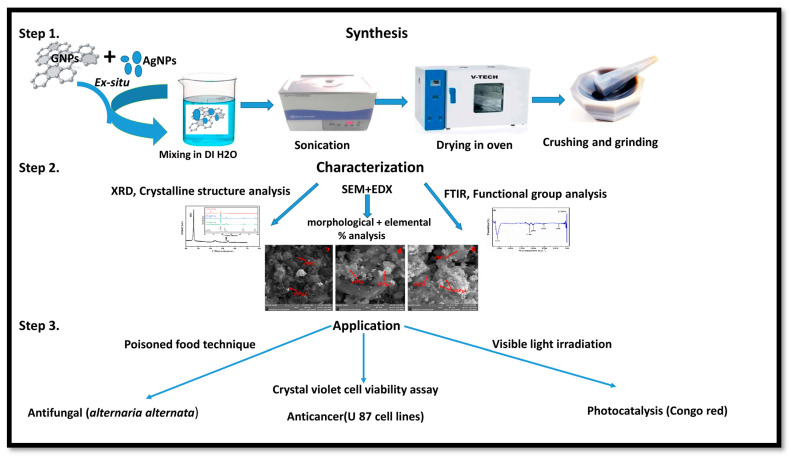
Schematic illustration of the whole study of (Ag)_1−x_(GNPs)_x_ nanocomposites.

**Table 1 molecules-28-04139-t001:** Inhibition of fungal growth diameter by (Ag)_1−x_(GNPs)_x_.

Incubated Days	Fungal Growth Inhibition (cm)				
	AgNPs	25% GNPs–Ag	50% GNPs–Ag	75% GNPs–Ag	Control
1	0	0	0	0	0
2	0.75	0	0	0	1.9
3	1.5	0.5	0.25	0.75	3.5
4	3.5	1	0.57	1.68	6.8
5	4.7	3.2	1.98	2.09	7.4
6	6	3.8	2.5	3.5	8.5
7	6	4	2.5	3.8	8.5

**Table 2 molecules-28-04139-t002:** Change in % degradation and reaction rate of CR with increasing time induced by (Ag)_1−x_(GNPs)_x_.

Nanomaterials	% Degradation	Rate Constant (min^−1^)
AgNPs	38.35	0.589
25% GNPs–Ag	59.12	0.849
50% GNPs–Ag	98.7	0.936
75% GNPs–Ag	76.07	0.860

## Data Availability

All the raw data of this research can be obtained from the corresponding authors upon reasonable request.

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
