# Peer review of "Fabrication and Characterization of Ag-Graphene Nanocomposites and Investigation of Their Cytotoxic, Antifungal and Photocatalytic Potential"

_molecules, 2023, doi:10.3390/molecules28104139_

Round 1
Reviewer 1 Report
The paper is written in a poor English and content is not always clearly exposed.
Notwithstanding there are some main point to be addressed:
Figure 5: What is the difference between growth inhibition and inhibitory effects?
Considering the irregularity of the mycelium spread, how diameters were measured?
The control of GNP is missing they may have inhibitory effects.
The pictures of Petri should have been taken vertically from above and not slanted.
Figure 7: When looking at the results one is surprised that Ag-NP have no toxic effect on cells, but on the contrary seems to protect them against the toxicity of GNP.
For this part it cannot be said that it is an anticancer activity but merely a cellular toxicity of the tested composites on a glioma cell line.
Figures 10 and 11: Ag-NP degrade Congo red. How it comes that this activity is the lowest in figure 10 and almost equivalent to the one of 50%Ag-NP/GNP composite in figure 11?
Is there a role of GNP alone ?
Figure 12: how authors know that Congo red is degraded until H2O and CO2?
For all the manipulations, how many time were they repeated? It would have been interesting to have a statistical analysis of results.
The name and the providers of products and reagents are not always indicated. It is also the case for the antifungal technique.
Considering these remarks which are only preliminary, although limited to the biological part, the paper is not acceptable as it is presented.
Author Response
Dear Reviewer,
Thank you so much for the positive comments, suggestions and proposed corrections. Thank you for giving me the opportunity to submit a revised draft of my manuscript. I appreciate the time and efforts that you and the reviewers have dedicated to providing your valuable feedback/ suggestions on my manuscript. I am grateful to the reviewers for their insightful comments on my paper as their comments have really further improved the structure and quality of our manuscript. I have been able to incorporate and respond to the changes provided by the reviewers. The manuscript has been thoroughly reviewed several time by all authors to further improve the structure and quality of manuscript. We have reviewed the manuscript multiple times to remove all typographic errors, major and minor mistakes. Now, we hope that the language of the manuscript is highly after editing and necessary changes.
I have highlighted the changes in red color within the manuscript.
Here is a point-by-point response to the reviewers’ comments and concerns.
- The paper is written in a poor English and content is not always clearly exposed.
Notwithstanding there is some main point to be addressed:
- Figure 5: What is the difference between growth inhibition and inhibitory effects?
Response: Growth inhibition means the inhibition in growth of fungi, while inhibitory effect described the role of nanocomposites on fungal growth with each passing day. The reason for measuring mycelial growth each day was to confirm the delayed growth process due to addition of Graphene.
- Considering the irregularity of the mycelium spread, how diameters were measured?
Response: The diameters were measured to the end corner of mycelia. All the experiments were conducted thrice and the average values were taken for graphs.
- The control of GNP is missing they may have inhibitory effects.
Response: The study was basically designed to check the incremental effect of GNPs on AgNPs. As mention in previous literature GNPs provide room to metallic nanoparticles to enhance the inhibition process. Alone monitoring the graphene effects was not the part of the current study.
- The pictures of Petri should have been taken vertically from above and not slanted.
Response: Thank you for your worthy suggestion, next time I will be care full while taking pictures. Your suggestion is highly appreciated.
- Figure 7: When looking at the results one is surprised that Ag-NP have no toxic effect on cells, but on the contrary seems to protect them against the toxicity of GNP.
Response: Silver nanoparticles cause toxicity, to overcome this GNPs were added. Silver /graphene nanocomposites showed improved results and maximum inhibition as graphene provides more room to hold Ag ions. (mentioned in manuscript)
For this part it cannot be said that it is an anticancer activity but merely a cellular toxicity of the tested composites on a glioma cell line.
Response: Thank you for your correction, yes it is cell cytotoxicity. Your deep comprehensive review/ suggestions are highly appreciated, it has really improved the structure and quality of the manuscript
- Figures 10 and 11: Ag-NP degrade Congo red. How it comes that this activity is the lowest in figure 10 and almost equivalent to the one of 50%Ag-NP/GNP composite in figure 11?
Response: Figure 10 shows the spectral absorption while Figure 11 mentions the reduction in concentration of Congo red. AgNPs have their different effect on dye as compared to 50%GNPs-Ag. (This phenomenon is also reported in various studies).
- Is there a role of GNP alone?
Response: Yes, alone GNPs have role, (mentioned in manuscript) but investigating alone GNPs was not part of study.
- For all the manipulations, how many time were they repeated? It would have been interesting to have a statistical analysis of results.
Response: I noted your suggestion, in future I will apply stat to my results. All experiments (biological part) were conducted in triplicate.
The name and the providers of products and reagents are not always indicated. It is also the case for the antifungal technique.
Response: Thank you for your suggestion. This comment has been addressed.
Considering these remarks which are only preliminary, although limited to the biological part, the paper is not acceptable as it is presented.
Reviewer 2 Report
The research article entitled “Fabrication and Characterization of Ag-Graphene Nanocomposites and Investigation of their Anticancer, Antifungal and Photocatalytic Potentials” focused on the preparations of Ag-Graphene Nanocomposites. They evaluated the antifungal and anticancer activity as well as Photocatalytic activity against CR for Ag-Graphene Nanocomposites and compared them with only silver nanoparticles. overall, the results are interesting, but key issues need to address like the confirmation assay for the formation of nanomaterials, the Reusability study for the photocatalyst, etc. There are many grammatical and sentence errors in the article, and the language organization needs to be improved. For these reasons, I conclude that the paper is not suitable for its current form and is recommended for publication with Minor revision.
-
Authors need to perform XRD and UV-VIs to confirm the formation of Ag-Graphene Nanocomposites as well as AgNPs.
-
The authors also need to calculate the MIC value have been determined.
-
Photocatalytic experiments: where is the scavenger test, to elucidate your mechanism?
-
Reusability study for the photocatalyst to understand the economic feasibility of the application.
-
Typographic errors need to be corrected. The language and grammar used throughout the manuscript need to be improved.
Author Response
Dear Reviewer,
Thank you so much for the positive comments, suggestions and proposed corrections. Thank you for giving me the opportunity to submit a revised draft of my manuscript. I appreciate the time and efforts that you and the reviewers have dedicated to providing your valuable feedback/ suggestions on my manuscript. I am grateful to the reviewers for their insightful comments on my paper as their comments have really further improved the structure and quality of our manuscript. I have been able to incorporate and respond to the changes provided by the reviewers. The manuscript has been thoroughly reviewed several time by all authors to further improve the structure and quality of manuscript. We have reviewed the manuscript multiple times to remove all typographic errors, major and minor mistakes. Now, we hope that the language of the manuscript is highly after editing and necessary changes.
I have highlighted the changes in red color within the manuscript.
Here is a point-by-point response to the reviewers’ comments and concerns.
- Authors need to perform XRD and UV-VIs to confirm the formation of Ag-Graphene Nanocomposites as well as AgNPs.
Response: Thank you for your suggestions. Manuscript has been updated with XRD and UV-Vis as suggested.
- The authors also need to calculate the MIC value have been determined.
Response: The study was designed to monitor the incremental effect of GNPs on AgNPs, the only single concentration 1mg/ml was taken. We are also working on some other antifungal projects, where we will be calculating the MIC values as well.
- Photocatalytic experiments: where is the scavenger test, to elucidate your mechanism?
Response: No further experiment was conducted, relate the results with previously reported findings.
- Reusability study for the photocatalyst to understand the economic feasibility of the application.
Response: Thank you so much for your quality comments. Unfortunately, due to limited resources, the reusability study was not achieved. In future projects we will take this comment into consideration from start. At the moment, we don’t resources in our Lab to perform this experiment. Hope you will cooperate.
- Typographic errors need to be corrected. The language and grammar used throughout the manuscript need to be improved.
Response: Thank you for your suggestion. The manuscript has been thoroughly reviewed several time by all authors to further improve the structure and quality of manuscript. We have reviewed the manuscript multiple times to remove all typographic errors, major and minor mistakes. Now, we hope that the language of the manuscript is highly after editing and necessary changes.
Round 2
Reviewer 1 Report
I acknowledge the almost complete revision of the text for the English.
Some editorial errors need to be addressed:
Line 82:Alternaria (italics)
Line 83: the name Alternaria alternata has to be abbreviated at its first apparition in the text, with the abbreviated name A. alternata between brackets, this one being used thereafter.
Line 716: Alternaria (italics)
Line 456: Escherichia coli and Staphylococcus aureus (not E. coli and S. aureus) in italics
Line 459-460: Candida albicans (not C. albicans) in italics
Line 466: Mucor racemosus (not M. racemosus) in italics
Line 232: Fusarium graminearum (not F. graminearum )
Line 561: Petri dishes (not petri plates)
Figure 6: The images are of poor quality and out of focus (besides the slanted photography) it does not permit to estimate the diameter sizes of mycelium growth.
Lines 505-506: why write the company names in uppercases? For all the purveyors of reagents and materials, the addresses are missing (city, country). Where the cell line obtained from?
Lines 32, 33, 227, 248, 293, 475, 476, 477, 574, 588, 592: mL (not ml), it is correct elsewhere in the text.
There are some more serious concerns:
Figure 4: scales ?
Figure 7: the vertical legend is not correct, it should be mycelium growth (and not mycelial growth inhibition as the greatest diameter of growth is with the control and it corresponds in the figure to 100% inhibition!). 7a is redundant with 7b (it seems to be the result at day 7) and may be omitted.
In 7a, why Alternaria alternata is indicated, it is useful? It should be altogether corrected, and written in italics.
Figure 9: the vertical scale indicates the contrary of what is meant as 100% inhibition corresponds to the control: % cell viability would be acceptable
How many times the different experiments have been repeated. It is said that they were done in triplicate but statistics have been done (figure 4?). An elementary statistic is mandatory (means and standard deviations) to know the quality of the results (repeatability), and to compare the different assays within the same experiment.
One cannot say the study with cancer cell line is the measure of an “anticancer effect.” It is merely a cytotoxic effect on cancerous cell line, likely of Ag-NP, and may differ from one cell line to the other (there is not another cell line for comparison). Thus the title of the paragraph has to be modified. Cytotoxicity of nanocomposites would be better.
Line 292-297: The sentences have to be rewritten. “The concentration of a dose” (? meaning) (Ag-NPs and Ag-GNPs nano-composites) “initially was given initially 3.15 μg/ml and increased up to 200 μg/ml.” It is simpler to write the range of concentration was from 3.15 μg/ml to 200 μg/ml.
As the concentration of nano composites increases the increased, “cell death increases means there is increased, indicating a decrease in percentage cell viability.” It is evident! One indication would suffice, e.g., cell viability.
The effect of GNPs-Ag is superior to the effect of Ag-NP alone, but do the authors have any explanation for the better activity of the sole 25% GNPs-Ag?
The IC50 can be calculated as, for example and according to the figure, the value for 25% GNPs-Ag is somewhere between 12.5 and 6.25 μg/mL. One would have expected an increase of activity with the concentration of Ag (or on the contrary about the same results with the three concentrations tested).
Author Response
Review Report (Round 2)
Dear editor and reviewer,
Thank you so much for reviewing our manuscript by providing your quality time, valuable comments/ recommendations. Your comments has really improved the quality and structure of our manuscript. Our team members have extensively and carefully reviewed the manuscript and properly addressed all suggestions/recommendations point by point as suggested by worthy reviewer. Further, all authors have extensively reviewed the English language of the manuscript and made extensive changes/ improvements wherever needed to further improve the structure, quality and grammar of the manuscript. Now, we hope that the revised manuscript is highly improved. We have again critically reviewed the manuscript for typos errors and other mistakes. All necessary changes made have been highlighted yellow. If you recommend further suggestions, let us know, we will be very happy to address.
Comments and Suggestions for Authors
I acknowledge the almost complete revision of the text for the English.
Some editorial errors need to be addressed.
Line 82: Alternaria (italics)
Response: The word Alternaria has been italicized throughout the manuscript. Thanks for your interest and suggestions to our manuscript.
Line 83: the name Alternaria alternata has to be abbreviated at its first apparition in the text, with the abbreviated name A. alternata between brackets, this one being used thereafter.
Response: This comment has been addressed as suggested. Thanks for correction.
Line 716: Alternaria (italics)
Response: The word Alternaria has been italicized throughout the manuscript. Thanks for your interest and suggestions to our manuscript.
Line 456: Escherichia coli and Staphylococcus aureus (not E. coli and S. aureus) in italics
Response: The E. coli and S. aureus has been updated with Escherichia coli and Staphylococcus aureus.
Line 459-460: Candida albicans (not C. albicans) in italics
Response: The C. albicans has been updated with Candida albicans and highlighted red in the revised manuscript.
Line 466: Mucor racemosus (not M. racemosus) in italics
Response: The M. racemosus has been updated with Mucor racemosus and highlighted red in the revised manuscript.
Line 232: Fusarium graminearum (not F. graminearum)
Response: The F. graminearum has been updated with Fusarium graminearum and highlighted red in the revised manuscript.
Line 561: Petri dishes (not petri plates)
Response: The “petri plates” has been replaced with “petri dishes” in the revised manuscript.
Lines 505-506: why write the company names in uppercases? For all the purveyors of reagents and materials, the addresses are missing (city, country). Where the cell line obtained from?
Response: The suggestion has been updated in the revised manuscript.
Lines 32, 33, 227, 248, 293, 475, 476, 477, 574, 588, 592: mL (not ml), it is correct elsewhere in the text.
Response: The word “ml” has been updated with “mL”. Thanks for correction.
There are some more serious concerns:
Figure 4: scales?
Response: As the nanoparticles (AgNPs and GNPs) were purchased from company. The SEM provided by company without scale on picture. Size was also confirmed by XRD.
Figure 7: The vertical legend is not correct, it should be mycelium growth (and not mycelial growth inhibition as the greatest diameter of growth is with the control and it corresponds in the figure to 100% inhibition!). 7a is redundant with 7b (it seems to be the result at day 7) and may be omitted.
Response: 7a has omitted.
In 7a why Alternaria alternata is indicated, it is useful? It should be altogether corrected, and written in italics.
Response: 7a has been omitted.
Figure 9: the vertical scale indicates the contrary of what is meant as 100% inhibition corresponds to the control: % cell viability would be acceptable
Response: Updated with % cell viability.
How many times the different experiments have been repeated. It is said that they were done in triplicate but statistics have been done (figure 4?). An elementary statistic is mandatory (means and standard deviations) to know the quality of the results (repeatability), and to compare the different assays within the same experiment.
One cannot say the study with cancer cell line is the measure of an “anticancer effect.” It is merely a cytotoxic effect on cancerous cell line, likely of Ag-NP, and may differ from one cell line to the other (there is not another cell line for comparison). Thus the title of the paragraph has to be modified. Cytotoxicity of nanocomposites would be better.
Response: Updated
Line 292-297: The sentences have to be rewritten. “The concentration of a dose” (? meaning) (Ag-NPs and Ag-GNPs nano-composites) “initially was given initially 3.15 μg/ml and increased up to 200 μg/ml.” It is simpler to write the range of concentration was from 3.15 μg/ml to 200 μg/ml.
As the concentration of nano composites increases the increased, “cell death increases means there is increased, indicating a decrease in percentage cell viability.” It is evident! One indication would suffice, e.g., cell viability.
Response: Lines Updated after rephrasing. Thanks for your correction and highlighting the sentence structure.
The effect of GNPs-Ag is superior to the effect of Ag-NP alone, but do the authors have any explanation for the better activity of the sole 25% GNPs-Ag?
Response: As described in manuscript the study circles around the effect of GNPs on AgNPs. Both these Nanoparticles show activity alone. With the addition of Grphene in Silver enhanced the activity.25 %GNPs-Ag showed better activity. But it is clear from the results that best was with 50%GNPs-Ag. Graphene has the ability to provide room to hold silver ions due to its electronic structure (described in earlier findings).
The IC50 can be calculated as, for example and according to the figure, the value for 25% GNPs-Ag is somewhere between 12.5 and 6.25 μg/mL. One would have expected an increase of activity with the concentration of Ag (or on the contrary about the same results with the three concentrations tested.
Response: Silver is an excellent candidate to assess anticancer activity. Silver together with graphene plays important role against cancer cell lines. Concentration of Silver nanoparticles (from lower to higher) effect the activity. As we add GNPs to AgNPs from 25 to 75%, at the same time the concentration of AgNPs reduces. AgNPs together with GNPs are reported with good results.
Kavinkumar, T., Varunkumar, K., Ravikumar, V., & Manivannan, S. (2017). Anticancer activity of graphene oxide-reduced graphene oxide-silver nanoparticle composites. Journal of colloid and interface science, 505, 1125-1133.
Motafeghi, F., Gerami, M., Mortazavi, P., Khayambashi, B., Ghassemi-Barghi, N., & Shokrzadeh, M. (2023). Green synthesis of silver nanoparticles, graphene, and silver-graphene nanocomposite using Melissa officinalis ethanolic extract: Anticancer effect on MCF-7 cell line. Iranian Journal of Basic Medical Sciences, 26(1).